# Role of Increasing Body Mass Index in Gut Barrier Dysfunction, Systemic Inflammation, and Metabolic Dysregulation in Obesity

**DOI:** 10.3390/nu17010072

**Published:** 2024-12-28

**Authors:** Fatima Maqoud, Francesco Maria Calabrese, Giuseppe Celano, Domenica Mallardi, Francesco Goscilo, Benedetta D’Attoma, Antonia Ignazzi, Michele Linsalata, Gabriele Bitetto, Martina Di Chito, Pasqua Letizia Pesole, Arianna Diciolla, Carmen Aurora Apa, Giovanni De Pergola, Gianluigi Giannelli, Maria De Angelis, Francesco Russo

**Affiliations:** 1Functional Gastrointestinal Disorders Research Group, National Institute of Gastroenterology IRCCS “Saverio de Bellis”, Castellana Grotte, 70013 Bari, Italy; fatima.maqoud@irccsdebellis.it (F.M.); domenica.mallardi@irccsdebellis.it (D.M.); francesco.goscilo@irccsdebellis.it (F.G.); benedetta.dattoma@irccsdebellis.it (B.D.); antonia.ignazzi@irccsdebellis.it (A.I.); michele.linsalata@irccsdebellis.it (M.L.); 2Department of Soil, Plant and Food Science, University of Bari Aldo Moro, 70126 Bari, Italy; giuseppe.celano@uniba.it (G.C.); carmen.apa@uniba.it (C.A.A.); maria.deangelis@uniba.it (M.D.A.); 3Center of Nutrition for the Research and the Care of Obesity and Metabolic Diseases, National Institute of Gastroenterology IRCCS “Saverio de Bellis”, Castellana Grotte, 70013 Bari, Italy; gabriele.bitetto@irccsdebellis.it (G.B.); martina.dichito@irccsdebellis.it (M.D.C.); giovanni.depergola@irccsdebellis.it (G.D.P.); 4Core Facility Biobank, National Institute of Gastroenterology IRCCS “Saverio de Bellis”, Castellana Grotte, 70013 Bari, Italy; letizia.pesole@irccsdebellis.it; 5Laboratory of Clinical Pathology, National Institute of Gastroenterology IRCCS “Saverio de Bellis”, Castellana Grotte, 70013 Bari, Italy; arianna.diciolla@irccsdebellis.it; 6Scientific Direction, National Institute of Gastroenterology IRCCS “Saverio de Bellis”, Castellana Grotte, 70013 Bari, Italy; gianluigi.giannelli@irccsdebellis.it

**Keywords:** obesity, body mass index (BMI), intestinal permeability, zonulin, inflammatory markers, fecal metabolomics, fecal microbiota

## Abstract

Aims: This study explores the link between body mass index (BMI), intestinal permeability, and associated changes in anthropometric and impedance parameters, lipid profiles, inflammatory markers, fecal metabolites, and gut microbiota taxa composition in participants having excessive body mass. Methods: A cohort of 58 obese individuals with comparable diet, age, and height was divided into three groups based on a priori clustering analyses that fit with BMI class ranges: Group I (25–29.9), Group II (30–39.9), and Group III (>40). Anthropometric and clinical parameters were assessed, including plasma C-reactive protein and cytokine profiles as inflammation markers. Intestinal permeability was measured using a multisaccharide assay, with fecal/serum zonulin and serum claudin-5 and claudin-15 levels. Fecal microbiota composition and metabolomic profiles were analyzed using a phylogenetic microarray and GC-MS techniques. Results: The statistical analyses of the clinical parameters were based on the full sample set, whereas a subset composed of 37 randomized patients was inspected for the GC/MS metabolite profiling of fecal specimens. An increase in potentially pro-inflammatory bacterial genera (e.g., *Slackia*, *Dorea*, *Granulicatella*) and a reduction in beneficial genera (e.g., *Adlercreutzia*, Clostridia UCG-014, *Roseburia*) were measured. The gas chromatography/mass spectrometry analysis of urine samples evidenced a statistically significant increase in m-cymen-8-ol, 1,3,5-Undecatriene, (E, Z) and a decreased concentration of p-cresol, carvone, p-cresol, and nonane. Conclusions: Together, these data demonstrated how an increased BMI led to significant changes in inflammatory markers, intestinal barrier metabolites, glucose metabolism, endocrine indicators, and fecal metabolomic profiles that can indicate a different metabolite production from gut microbiota. Our findings suggest that targeting intestinal permeability may offer a therapeutic approach to prevent and manage obesity and related metabolic complications, reinforcing the link between gut barrier function and obesity.

## 1. Introduction

The obesity pandemic is a severe social and health issue that is rapidly and continuously growing, particularly in middle- and high-income countries. In Italy, the total cost of managing obesity was estimated at EUR 13.34 billion, according to 2020 statistics, with 59% attributed to direct healthcare costs and 41% to indirect costs [1]. Beyond the financial burden, obesity has a significant social impact, affecting education, employment, and quality of life while also increasing the risk of morbidity and mortality.

Obesity is not merely a condition related to energy imbalance; it is a complex pathology with multifactorial causes, including genetic, environmental, and behavioral factors. However, the genetic component contributes only a small percentage (approximately 2%) to obesity [2]. This suggests that the primary causes of obesity are environmental, with lifestyle factors playing a significant role in moderating the effects of a genetic predisposition to obesity at the individual level [3].

The body mass index (BMI), which relates weight (in kg) to height (expressed in kg/m^2^), remains the primary parameter used to classify obesity. In line with the World Health Organization, four categories of increasing risk have been identified: overweight (25–29.9 kg/m^2^), first grade of obesity (30–34.9 kg/m^2^), second grade of obesity (35–39.9 kg/m^2^), and third grade of obesity (>40 kg/m^2^).

The risk of developing comorbidities is directly proportional to an increase in BMI, which is caused by hyperplasia and hypertrophy of visceral adipose tissue (VAT) [4]. This tissue increases intra-abdominal pressure and elicits the release of inflammatory cytokines and chemokines, leading to a pro-inflammatory state and oxidative stress in obese individuals. Additionally, obesity is associated with low-grade systemic inflammation [5], contributing to the development of various comorbidities, including metabolic dysfunction [6] and gastrointestinal (GI) dysfunction.

At the intestinal level, the accumulation of adipose tissue and local inflammation disrupt both physical and functional components that are part of the intestinal barrier [7]. This alteration affects in turn the intestinal microbiome, the nervous system, the immune system, and electrolyte and macromolecule exchange too [8]. Over time, this process compromises the intestinal barrier’s integrity by altering the adhesion between enterocytes, affecting tight junction (TJ) proteins—key regulators of homeostasis in epithelial and endothelial tissues [9].

The loss of intestinal barrier integrity and an increased intestinal permeability (IP) are now recognized as complications associated with obesity. The intestinal barrier acts as a dynamic unit that facilitates nutrient absorption while limiting the passage of harmful or unwanted molecules into the body [10]. The barrier’s functionality can be assessed through non-invasive methods such as a sugar absorption test (SAT); this involves orally administering probes such as lactulose (L), mannitol (M), sucrose, and sucralose (S) in a fasting state, followed by measuring their recovery in urine over specific time intervals (5–6 h) [11]. When the barrier is compromised, these probes are absorbed more, and their elevated presence in urine indicates impaired intestinal integrity [11,12]. Additionally, serum analysis of fatty-acid-binding proteins (I-FABPs) and diamine oxidase (DAO) can help in evaluating gastrointestinal barrier integrity [11]. High levels of I-FABPs indicate impaired transcellular permeability, while higher values of DAO represent increased cellular damage [13]. Serum and fecal zonulin levels are also considered as reliable markers of intestinal permeability [14,15].

The gut microbiota is increasingly recognized as a key factor related to the development of obesity [16]. Several studies have found a strict association between obesity and an imbalance among the commensal proportion of the gut ecosystem. This imbalance in the microbial community can compromise the intestinal barrier’s integrity and gut-associated lymphoid tissue (GALT) [17], enabling the translocation of harmful molecules like lipopolysaccharides (LPSs) and bacterial components that activate inflammatory pathways both at the local and systemic level [18,19]. Additionally, evidence shows that dysbiosis can interfere with the release of satiety-regulating peptides, leptin sensitivity [20,21], and cholecystokinin signaling via the vagus nerve, ultimately affecting appetite control and food intake [22].

In the current management of obese patients, BMI is primarily used to classify individuals based on their anthropometric characteristics. However, it should be considered alongside other factors, such as gastrointestinal barrier integrity and local and systemic inflammation, to improve treatment strategies. It is hypothesized that the weakening of the intestinal barrier may exacerbate the clinical condition of obesity. As obesity progresses, the intestinal barrier damage increases, including alterations in the microbiota, its production in terms of volatile organic compound metabolites into the lumen and evidently rescued from fecal specimens. Based on these premises, our approach aims at investigating how intestinal permeability, microbiota composition, fecal metabolites, and local and systemic inflammation evolve dynamically as BMI increases. Thus, the interaction between intestinal health and obesity would suggest how the maintenance or the restoring of intestinal barrier function may help in attenuating the metabolic and inflammatory effects of obesity.

## 2. Materials and Methods

### 2.1. Population Cohort and Clinical Trials

The study took place at the Italian Nutritional Center for Research and Treatment of Obesity and Metabolic Diseases, part of the National Institute of Gastroenterology IRCCS “Saverio de Bellis” in Castellana Grotte (BA), from April to November 2022. Informed written consent was obtained from all participants involved in this study. The study protocol (Prot. n. 170/CE De Bellis) was approved by the internal Medical Ethics Committee and adhered to the Declaration of Helsinki guidelines (1964). Additionally, it was registered on ClinicalTrials.gov under the identifier NCT05477212. The research, a longitudinal study in the already explored cohort, was focused on participants with excessive body mass, i.e., with BMI over 25 kg/m^2^, aged from 18 to 65 years. Participants underwent several examinations, including medical history reviews, smoking and alcohol consumption habits, physical examinations, and laboratory testing. A cohort of 58 patients not suffering from pathologic conditions including irritable bowel syndrome (IBS), inflammatory bowel disease, celiac disease, gastrointestinal malignancies, constipation, or gallstones was selected. Based on the statistical analysis of clinical/anthropometric variables, the cohort was divided into three sub-groups: Group I (BMI 25–29.9), Group II (BMI 30–39.9), and Group III (BMI > 40).

Detailed anthropometric and bioelectrical impedance analyses were performed at recruitment, and fasting blood, urine, and stool samples were collected. A sugar absorption test (SAT) was administered to evaluate gastrointestinal permeability. In addition to having filled in the written informed consent, participants met the following inclusion criteria: age between 18 and 65 years and a BMI greater than 25 kg/m^2^. Exclusion criteria included the presence of acute or chronic diseases such as infections, eating disorders, liver failure, cardiovascular and cerebrovascular diseases, respiratory failure, renal disease, type 1 diabetes, acute gastrointestinal conditions, smoking, or use of medications that could affect gut health (e.g., antibiotics, NSAIDs, and immunosuppressants) within three months before the study. Pregnant or breastfeeding women, frail elderly individuals, and those with psychiatric disorders, severe mental illness, or substance abuse issues were also excluded from the study.

### 2.2. Anthropometric Assessment

BMI was computed based on height and weight measurements. These were taken by means of a stadiometer and a calibrated scale, with patients fasting and their bladders emptied. In the case of waist circumference (WC), patients were asked to stand upright with their abdomen exposed and weight evenly distributed on both legs. The position between the rib lower edge and the iliac crest (approximately at the navel level) was used to tape the measurement.

Both the measurements are based on the guidelines reported by WHO (https://iris.who.int/bitstream/handle/10665/44583/9789241501491_eng.pdf accessed on 15 October 2024).

Participants with excessive body mass were categorized as follows: overweight (25–29.9 kg/m^2^), first obesity grade (30–34.9 kg/m^2^), second obesity grade (35–39.9 kg/m^2^), and third obesity grade (>40 kg/m^2^). A detailed description of the clustering analysis performed a priori is provided in the following section.

### 2.3. Bioelectrical Impedance Analysis (BIA)

Body composition was assessed by means of single-frequency bioimpedance (Akern Bioresearch, Florence, Italy), with measurements conducted by dietitians following standard procedures. All patients were examined according to the European Society of Parenteral and Enteral Nutrition (ESPEN) recommendations (https://www.gavecelt.it/nuovo/links-utili/espen-european-society-parenteral-and-enteral-nutrition accessed on 15 October 2024). Patients fasted for 12 h and refrained from consuming alcohol or engaging in physical exercise in the 24 h prior to the examination. The assessment was performed with the patient in a supine position, legs apart, after removing shoes and socks and cleaning the electrode application sites (BIATRODES Akern, Florence, Italy) with alcohol. A previous study thoroughly described the electrode placement points [23]. Using the patient data of weight, height, sex, and age, the Akern software (BODYGRAM^R^ PLUS version 2019) analyzed the two vector impedance variables: Resistance (Rz) and Reactance (Xc). The software then calculated the Phase Angle (PA), derived from their ratio. It estimated the Fat Mass (FM), Fat-Free Mass (FFM), Metabolically Active Cellular Mass (BCM), Total Body Water (TBW), and Extracellular Body Water (EBW) using predictive equations.

### 2.4. Clinical Biochemistry

After overnight fasting, blood samples were collected early morning (between 8 a.m. and 9 a.m.). Glucose concentrations in the fasting plasma were determined by using the glucose oxidase method (Hitachi, Boehringer, Mannheim, Germany), while glycated hemoglobin (HbA1c) was analyzed by means of an Architect c8000 chemical analyzer (Abbott, Chicago, IL, USA). Serum insulin concentrations were measured using a radioimmunoassay method (Behring, Scoppito, Italy), and insulin resistance was calculated using the formula of Homeostasis Model Assessment-Insulin Resistance (HOMA-IR): (fasting insulin × fasting glucose)/405, based on a normal range between 0.23 and 0.25. We profiled the whole lipid content by means of the automated colorimetric method (Sclavus, Siena, Italy) that allowed for measuring the levels of total cholesterol, HDL cholesterol, and triglycerides, whereas LDL (low-density lipoprotein) and cholesterol were calculated using the Friedewald equation. Uric acid levels were determined using the URICASE/POD method (Boehringer, Mannheim, Germany), and levels of aminotransferase, γ-glutamyl transpeptidase (GT), and creatinine was measured using an automated system (UniCel Integrated Workstations DxC 660i, Beckman Coulter, Fullerton, CA, USA). Serum ferritin levels were measured with Access Ferritin Reagent Packs (Beckman-Coulter AB, Bromma, Sweden), while 25(OH) vitamin D was analyzed using a chemiluminescence method (Diasorin Inc., Stillwater, OK, USA).

### 2.5. Inflammation Marker Evaluation

Quantitative measurements of the levels of circulating interleukin-6 (IL-6), interleukin-8 (IL-8), interleukin-10 (IL-10), and tumor necrosis factor-alpha (TNF-α) were detected using ELISA kits from Elabscience, Houston, TX, USA (E-EL-H0102, E-EL-H6008, E-EL-H01103, E-EL-H0109). Lipopolysaccharide (LPS) concentration was quantified using an ELISA kit from Cloud-Clone Corp., Katy, TX, USA (SEB526GE).

### 2.6. Intestinal Permeability Evaluation

A sugar absorption test (SAT) was performed to assess gastrointestinal permeability. On the recruitment day, urine was collected at baseline (time 0) before sugar intake, and endogenous sugar levels were evaluated. Patients then ingested a solution containing lactulose (10 g), mannitol (5 g), and sucrose (40 g) dissolved in 100 mL of water. Urine samples were collected for 5 h following the ingestion of the solution. For sample preservation, 500 µL of 20% (*w*/*v*) chlorhexidine was added to each collection, independently of the final volume. The total urine excreted volume was detected. A total of 2 mL sample from each collection was extracted and stored at −80 °C until analysis. Chromatographic analysis was performed to quantify the three sugar probes (lactulose, mannitol, and sucrose) in urine samples collected at baseline and 5 h after sugar administration, following the previously described methodology [24]. The lactulose/mannitol ratio was used to determine intestinal permeability (IP). A ratio equal to or greater than 0.030 was considered the threshold for altered small intestinal permeability (IP-s).

### 2.7. Gastrointestinal Barrier Integrity Evaluation

Levels of zonulin both in serum and feces were assessed using ELISA kits from Immunodiagnostik AG, Bensheim, Germany (K5601, K5600). Serum and fecal levels under 48 ng/mL and 107 ng/mL were considered within the normal range for zonulin levels, as indicated in the manufacturer’s instructions. Additionally, intestinal fatty-acid-binding protein (I-FABP) serum levels were assessed using ELISA kits from Thermo Fisher Scientific, Waltham, MA, USA (EHFABP-2). Urine samples were used to measure indican and skatole levels with a colorimetric assay kit (KA3944, Indican Assay Kit, ABNova Corporation, Taipei, Taiwan) and Thermo Scientific’s Dionex high-performance liquid chromatography (HPLC) equipment, respectively. Indican levels above 20 mg/L were used to assess a status of fermentative dysbiosis, whereas skatole levels exceeding 40 µg/L suggest putrefactive dysbiosis. Serum levels of claudin-5 and claudin-15 were assessed using MyBioSource ELISA kits from MyBioSource, San Diego, CA, USA (MBS094541, MBS161669).

### 2.8. Fecal and Urinary VOC Profiles from Untargeted Metabolomics (GC/MS)

Gas chromatography/mass spectrometry volatile organic compounds (VOCs) were profiled twice and specifically before and after VLCKD treatment both for urine and fecal specimens.

Concerning fecal specimens, GC/MS metabolomic analysis was performed as already previously described based on an aliquot of 1 g from each fecal sample [25].

Briefly, the untargeted analysis was run on a Clarus 680 (Perkien Elmer, Waltham, MA, USA) equipped with a Rtx-WAX capillary column (30 m × 0.25 mm i.d., 0.25 μm film thickness) (Restek, Bellfonte, PA, USA). The gas chromatography system was coupled with a single quadrupole mass spectrometer Clarus SQ 8C.

Compound identification relied on matching the obtained spectra against the National Institute of Standards and Technology (NIST) 2008 library, using a match score threshold of >85% and a peak area greater than 1,000,000. Additionally, the Automated Mass Spectral Deconvolution and Identification System (AMDIS) was used as an effective tool for analyte identification.

### 2.9. Fecal DNA Extraction and 16S Metataxonomics

Total DNA was extracted from 39 patient stool samples with the QIAamp FAST DNA Stool Mini Kit (Qiagen, Hilden, Germany). DNA concentration was measured by means of a Qubit Fluorometer 1.0 (Invitrogen Co., Carlsbad, CA, USA). 16S targeted gene sequencing was performed at Genomix4life S.R.L. company (Baronissi, Salerno, Italy) based on the universal 16S primer couple (Illumina, San Diego, CA, USA). To avoid contamination, we added a negative control to the experiment. The pooled samples were subjected to cluster generation and sequenced on the MiSeq platform (Illumina, San Diego, CA, USA) in a 2 × 300 paired-end format.

After sequencing, the bioinformatics protocol included firstly the estimate of the raw reads’ quality by means of FastQC software version 0.12.1 (https://www.bioinformatics.babraham.ac.uk/projects/fastqc/, accessed on 15 October 2024). The QIIME2 microbiome platform (version 2020.8) and nested plugins were used in the denoising and taxonomic assignment steps. SILVA 138 (https://www.arb-silva.de/documentation/release-138/, accessed on 15 October 2024) was the database used to retrieve taxa annotations and relative abundances.

### 2.10. Statistical Analyses

Sample clusters were a priori inspected based on Discriminant Analysis of Principal Components (DAPC). The fitting between “a priori” obtained clusters (based on the minimum of the Bayesian Information Curve—BIC) and the groups assigned “a posterior” was used to assess the reliability of the classification. Specifically, the find.cluster function within the DAPC R package was used to assess the number of clusters detectable without any assumption of sample belonging. The fitting between the “a priori” and the “a posterior” analyses is reported in a DAPC R “assign plot” where the two assignments for each sample have been compared.

Categorical independent variables were compared using a two-way ANOVA/one-way ANOVA test followed by a Tukey’s post hoc multiple test correction.

Pairwise comparisons were performed using a Welch’s BH corrected test and graphically rendered as volcano plots where increased and decreased variables were reported. Correlations between variables were obtained using a Pearson’s test. Only statistically significant correlations with an r value greater than 0.6 were maintained.

## 3. Results

### 3.1. Cohort Description

Fifty-eight participants were enrolled, and clinical and anthropometric parameters were comprehensively evaluated. Clinical and biochemical parameters were measured on the complete patient set.

A subset of 37 patients underwent randomization for fecal and urinary metabolomic analysis, along with 16S rRNA gene sequencing. Participants were subsequently categorized into subgroups based on the World Health Organization BMI classification.

### 3.2. “A Priori” Clustering Analysis

Due to the robustness of clinical data in differentiating patient subgroups, the entire sample batch was initially analyzed using an a priori DAPC clustering approach. As reported in Figure 1, the find.cluster R function resulted in a possible stratification of samples resumed in the Bayesian Information Criterion (BIC) curve that identified four distinct clusters (as indicated by the minimum point of the broken curve). These clusters were spatially separated, clusters 2 and 3 almost overlapping in the second quadrant (Figure 1). The a posterior DAPC based on the known group assignments perfectly reflects the same spatial splitting (Figure 2). The matching of assignments between the a priori and a posterior DAPC was found for each one of the analyzed samples. A great separation between the overweight and third obesity patient class emerged, whereas patients in the first and second obesity classes showed relatively fewer differences.

Examining those variables with a greater impact on DAPC clustering, we identified a list that includes ferritin, serum zonulin, waist circumference, serotonin, insulin, interleukin-8, claudins, ghrelin, BMI, obestatin, and others (Figure 3A). As shown in Figure 3B, the fit demonstrates a perfect match between the a priori and a posterior assignments.

In other words, proportions of the successful reassignments have been collapsed in each cell: heat colors represent membership probabilities (red = 1, white = 0), and blue crosses represent the DAPC a priori cluster. Based on this evidence, which comes from the reliability of clinical/biochemical measurements, we merged the 1st and 2nd groups and compared this against overweight and obese patients belonging to class III (obe3). Thus, although the assignment reveals four different groups, we aimed to maximize the difference of the subsequently analyzed statistics using the abovementioned three groups.

### 3.3. Demographic, Anthropometric, and Bioimpedance Characteristics

Table 1 shows the demographic, anthropometric, and bioimpedance comparisons among participants divided into three BMI categories: Group I (BMI 25–29.9), Group II (BMI 30–39.9), and Group III (BMI > 40). Overall, the groups have no significant differences in mean age or height. The mean ages for Groups I, II, and III were 45.75 ± 4.63, 43.43 ± 3.33, and 37.75 ± 3.92, respectively, while the mean heights were 170.3 ± 2.86, 168.9 ± 1.87, and 165.4 ± 5.61, respectively. However, as expected, statistically significant differences (*p* < 0.05) were found in weight and waist circumference, with weight increasing progressively from 78.44 ± 2.77 in Group I to 94.84 ± 2.88 in Group II and 118.6 ± 9.29 in Group III. Similarly, waist circumference increased from 95.63 ± 2.43 in Group I to 109.5 ± 4.12 in Group II and 128.3 ± 4.1 in Group III. Bioimpedance analysis revealed statistically significant differences in fat and fat-free mass across the groups. Additionally, total body and extracellular body water significantly increased, particularly in Group III, where total body water averaged 28.63 ± 4.12, compared to 16.91 ± 1.93 in Group I and 23.03 ± 8.93 in Group II.

### 3.4. Serum Biochemistry Analysis

As shown in Table 2, the ANOVA revealed no significant differences among the three groups for total cholesterol, HDL, LDL, uric acid, iron, ferritin, and creatinine levels, indicating an overall homogeneity in these parameters. However, a marked statistical significance (*p* < 0.001) was observed for triglyceride levels, with Group III exhibiting an increase of 196.72% and 131.46% relative to Groups I and II, respectively. Additionally, 25-OH-vitamin D levels showed substantial reductions, with Group III showing a decrease of 42.08% and 39.19% compared to Groups I and II, respectively.

### 3.5. Glucose Metabolism and Endocrine Hormone Level

Dealing with glucose metabolism, Figure 4 shows that Group III, which includes individuals affected by severe obesity, had a notable decline in glucose metabolism compared to Groups I and II (Figure 4a). As BMI increased, blood glucose levels rose significantly in Groups II and III compared to Group I (Figure 4a). In Group III, this increase was joined with a substantial increase in insulin levels (Figure 4b), leading to elevated HOMA-IR values (Figure 4c). Additionally, obestatin levels significantly decreased in Groups II and III compared to Group I (Figure 4d). In contrast, ghrelin levels increased significantly in these groups compared to Group I (Figure 4e).

### 3.6. Levels of Inflammatory Factors

PCR levels showed a marked increase in Group III compared to Group I, with recorded values of 0.163 ± 0.09 pg/mL, 0.41 ± 0.31 pg/mL, and 0.83 ± 0.62 pg/mL for Groups I, II, and III, respectively (Figure 5). Additionally, Group III exhibited a significant increase in pro-inflammatory markers, including IL-6, with 30.59 ± 10.19 pg/mL, compared to the 5.35 ± 1.7 pg/mL and 20.38 ± 4.6 pg/mL detected in Groups I and II, respectively. Conversely, a notable decrease in anti-inflammatory markers, i.e., IL-10, was observed in both Groups II and III compared to Group I. Concerning TNF-α and IL-8 levels, no significant changes were detected (Figure 5).

### 3.7. Intestinal Barrier Function and Integrity

The altered intestinal permeability, measured in terms of the lactulose/mannitol (Lat/Man) absorption ratio, showed a progressive and significant increase across BMI groups (Figure 6). Significant differences were found in Groups II and III. Regarding intestinal barrier integrity, serum I-FABP concentrations were significantly higher in Groups II and III than in Group I. Another marker, claudin-5, displayed a notable decrease, with Group III showing approximately a 50.12% reduction compared to Group I (Figure 6). No significant differences were found in the fecal and serum levels of zonulin and claudin-15 across groups.

Additionally, a linear regression analysis evaluating the relationship between BMI and the intestinal permeability marker I-FABP is reported (Figure 7). The regression coefficient (β) for BMI and I-FABP was significantly higher in Group II (β = 0.21156, *p* = 0.011) and Group III (β = 0.30897, *p* < 0.001) compared to Group I (β = 0.04108, *p* = 0.64), indicating a stronger association in the higher BMI groups.

### 3.8. Intestinal Dysbiosis and Bacterial Translocation

Figure 8 shows that urinary indican levels significantly increased with an increase in BMI, with values of 50.36 ± 6.76 mg/L, 53.64 ± 3.69 mg/L, and 75.83 ± 6.74 mg/L in Groups I, II, and III, respectively. Similarly, circulating LPS levels were significantly elevated in Group III compared to Group I. No significant differences were observed in skatole levels across groups, with urinary skatole concentrations consistently below 20 μg/L. In contrast, indican levels exceeded 20 mg/L in all groups, indicating a dysbiosis characterized by a predominance of fermentation.

### 3.9. Statistically Different Urinary VOCs

Based on the BMI stratification, three groups were pairwise compared. Looking at urinary VOCs, the volcano plots highlight a statistically significant increased concentration of m-cymen-8-ol and 1,3,5-Undecatriene, (E, Z) and a decreased concentration of p-cresol in Group II versus Group I (Figure 9). At the same, obese patients belonging to Group III showed a decreased concentration in carvone, p-cresol, and nonane when compared with overweight patients.

### 3.10. Metataxonomic Differences Based on Clustering-Based BMI

When microbiota taxa at the genus level were inspected in terms of pairwise comparisons, the set of statistically significant variables included *Akkermansia*, *Adlercreuzia*, and *Family XIII AD 3011 group* (belonging to the Anaerovoracaceae family), which were decreased in Group II compared to Group I samples. On the other hand, when samples classified in Group III were compared to Group II, *Libanicoccus* and *Family XIII AD 3011 group* increased. In contrast, *Monoglobus*, *Clostridium sensu stricto 1* and unassigned genera resolved at the higher taxonomic level—family of Oscillospiraceae decreased (Figure 10).

*Roseburia, Clostridia UCG-014*, and *Adlercreuzia* decreased in Group III compared to Group I, while *Dorea*, *Howardella*, *Granulicatella*, *Slackia, Senegalimassilia*, *Holdemanella*, *Libanicoccus*, and *Catenibacterium* genera increased (Figure 10).

### 3.11. Statistically Different Fecal VOCs

Statistical significant fecal metabolites from 37 samples, inspected by fold change analysis, have been reported in Table 3.

After merging together all statistically significant variables into a single matrix describing the sample set (Table 3 for fecal VOC), a correlation analysis was conducted to determine whether specific inter-group variables could collectively explain a phenotype. Beyond the necessity of elucidating a biological rationale, it is noteworthy from a statistical perspective that ferritin exhibited positive correlations with *Holdemanella, Howardella*, and *Oscillospiraceae*, as well as with insulin and *Libanicoccus*. Additionally, inter-group comparisons of VOC taxa revealed several positive correlations, including *Adlercreutzia* with 1H-indole, 5-methyl-, and *Clostridia UCG-014* with 2-Undecene, 6-methyl-, (Z)- and methanethiol (Figure 11).

## 4. Discussion

Obesity and its progression leads to multiorgan and tissue dysfunction and poses a significant public health challenge [26]. Despite its widespread impact, due to an incomplete understanding of the disease’s underlying mechanisms, there are no safe, accessible, or sustainable treatments. In clinical practice, obesity is usually classified using BMI [27], a formula that categorizes obese individuals into four groups based on two anthropometric values: weight and height [27]. Consequently, it only assesses the ratio of these two variables without considering the amount of subcutaneous and visceral fat and, therefore, the inflammatory status of the patients [28]. Chronic low-grade inflammation is increasingly recognized as a local and systemic factor in obesity-related dysfunction [6]. Emerging research highlights the loss of intestinal integrity, mainly based on the disruption of tight junction complexes (ZO-1, occludin, and claudin), as a key but underexplored source of systemic inflammation [29,30]. This disruption leads to increased intestinal permeability, interrupting the exchange of nutrients and electrolytes and allowing the uncontrolled passage of pro-inflammatory agents, metabolites, and microbes into the bloodstream. As a result, immune responses are activated locally and systemically in the gut, further exacerbating inflammation [31]. Although the link between intestinal permeability and obesity is well known, the exact mechanisms remain poorly understood, hindering effective prevention and treatment strategies.

Additionally, increasing evidence underscores the critical role of the gut microbiota in the development and progression of obesity [31]. The microbiota, often described as a “metabolic organ”, is vital for nutrient processing, energy balance, hormonal regulation, and body weight control [32,33]. The combination of microbiota imbalances and a compromised intestinal barrier intensifies metabolic endotoxemia, contributing to chronic low-grade inflammation and exacerbating obesity-related conditions such as non-alcoholic fatty liver disease and type 2 diabetes. Recent studies have demonstrated how metabolites produced by the gut microbiota can influence the expression of key tight junction proteins like Zo1, thereby impacting intestinal permeability [34]. Specific metabolites and conjugated fatty acids have been shown to promote Zo1 expression and improve barrier function, suggesting potential therapeutic avenues that merit further investigation [35,36,37].

As expected, patients belonging to the group with highest BMI showed significant differences in weight and waist circumference. Significant variations were also observed in Fat-Free Mass, Fat Mass, Total Body Water, and Extracellular Body Water. However, blood values such as Total Cholesterol, HDL, LDL, uric acid, iron, ferritin, and creatinine showed no significant differences between groups. Conversely, triglycerides were notably higher in Group III than in Groups I and II. Additionally, 25-OH vitamin D levels were lower in Group II than in Group I, likely due to the higher uptake of vitamin D by adipose tissue, reducing its circulation and bioavailability [38].

Glucose metabolism indicators, including fasting glucose, insulin levels, and the insulin resistance index HOMA-IR, worsened as BMI increased [39]. Glucose and insulin levels rose in Groups I and III compared to Group I, while HOMA-IR was highest in Group III. Appetite-regulating hormones like obestatin and ghrelin were also elevated in Groups II and III compared to Group I.

Among our results, and in line with the literature [4,13,40], specific inflammatory markers showed differences in Group III, as in the case of pro-inflammatory markers including PCR and IL-6 and lower levels of the anti-inflammatory marker IL-10. TNF and IL-8 levels, however, remained similar across the groups. Gut barrier function and integrity markers, such as the lactulose/mannitol absorption ratio, increased significantly in Groups II and III compared to Group I and were also higher in Group III compared to Group II. Serum I-FABP levels followed a similar trend, while claudin-5 levels progressively decreased as BMI increased. No significant differences were observed in serum and fecal zonulin or claudin-15 levels. Emerging evidence underscores that obesity-associated intestinal barrier dysfunction is influenced by factors beyond nutrient intake and tissue impairment, traditionally linked to tight junction (TJ) downregulation. Key contributors such as lifestyle choices, physical activity, dietary patterns, and notably, dysbiosis have been identified as pivotal modulators of intestinal barrier integrity. These insights point to the potential for innovative, multifaceted strategies to enhance barrier function, offering new therapeutic avenues for managing obesity and its related complications [40,41,42].

Our investigation, based on VOC analysis, revealed a reduction in the lipid metabolites (e.g., tetradecane, squalene, 9-octadecenoic acid (Z)-methyl ester) involved in lipid metabolism and the synthesis of antioxidant molecules. Additionally, as BMI increases, there was a notable reduction in aromatic and volatile metabolites (e.g., benzenepropanol, phenol 3-methyl-5-(1-methylethyl)); this reduction may reflect a decreased bacterial metabolic activity in the catabolism of aromatic amino acids or other microbial pathways involved in the production of volatile compounds, such as indole, which have protective and anti-inflammatory effects on the intestinal barrier. Considering these findings alongside the extensive scientific evidence from multiple studies, it is evident that a metabolomic analysis of the volatile organic compounds (VOCs) in biological materials offers valuable functional insights for the clinical diagnosis and monitoring of a wide range of diseases, including obesity. Consistent with the existing literature, our results indicate that obese individuals exhibit distinct alterations in volatile metabolites, which are frequently associated with processes such as lipid peroxidation, protein oxidation, and dysregulated intestinal microbial metabolism [43,44]. Additionally, our patient cohort analysis revealed that an increasing BMI correlates with compromised intestinal barrier integrity and VOC alterations linked to antioxidant molecule synthesis. These findings suggest an increase in oxidative stress as a consequential mechanism [45,46].

Particular attention was given to the reduction in sulfur-containing metabolites and indoles (e.g., methanethiol, disulfide dimethyl, 1H-indole-5-methyl), which are often derived from the metabolism of tryptophan or sulfur-containing compounds and possess anti-inflammatory properties, contributing to the regulation of intestinal integrity. Their reduction also indicates a decline in the population of bacteria producing these beneficial compounds, such as *Adlercreutzia* or other producers of indoles and short-chain fatty acids (SCFAs), which are crucial for maintaining intestinal mucosal health [47]. In agreement with the existing literature, including the findings by Alkhouri et al. [48], compounds such as isoprene, 1-decene, 1-octene, ammonia, and hydrogen sulfide have been shown to be significantly elevated in individuals with obesity compared to their lean counterparts. Our results align with these observations, further substantiating these metabolic distinctions.

Microbiota analysis revealed an increase in potentially pro-inflammatory bacterial genera (e.g., *Slackia*, *Dorea*, *Granulicatella*) and a reduction in beneficial genera (e.g., *Adlercreutzia*, Clostridia UCG-014, *Roseburia*), suggesting a dysbiosis associated with increased BMI. In particular, *Adlercreutzia* reduces hydrogen levels in the intestinal lumen, fostering a healthy environment for other fiber-fermenting bacteria [47,49]. Its reduction can lead to hydrogen accumulation, impairing microbial fermentation efficiency and reducing SCFA production, such as butyrate [50].

The intestinal dysbiosis we observed in obese subjects, characterized by increased pro-inflammatory bacteria and reduced beneficial bacteria (such as *Roseburia* and *Adlercreutzia*), results in the diminished production of protective metabolites like butyrate and indole compounds [51].

The reduced butyrate production compromises the integrity of the intestinal barrier, increasing intestinal permeability and allowing endotoxins (LPSs) to enter the bloodstream, triggering a systemic inflammatory response [51]. Chronic inflammation, in turn, exacerbates insulin resistance, alters lipid metabolism, and disrupts hormonal regulation (ghrelin/obestatin) [52], creating a vicious cycle that perpetuates BMI increases and worsens the metabolic conditions associated with obesity [53].

This study presents some limitations related to the non-homogeneous number of randomized samples used in the parallel and concurrent analyses. The major strength relies on the possibility of matching statistical a priori group predictions, based on biochemical/anthropometric variables, with an a posterior group assignment based on BMI stratification. In the near future, a major effort will be made to link bacterial taxa and metabolites by means of transcribed gene profiles.

## 5. Conclusions

Our findings emphasize that gut permeability worsens as BMI increases. This leads to systemic inflammation, which exacerbates obesity-related metabolic disorders. A higher BMI disrupts the gut barrier and microbiota composition, allowing pro-inflammatory agents to enter the bloodstream, triggering chronic inflammation. This inflammatory state further contributes to insulin resistance, altered lipid metabolism, and hormonal imbalances. Our data suggest that therapeutic strategies aimed at restoring gut barrier integrity and modulating the gut microbiota may offer promising avenues for mitigating the metabolic complications of obesity. The clinical data evidently allowed to a priori cluster individuals with an excess of body weight, and surprisingly these groups coincide with those based on BMI ranges. The metabolomic and metataxonomic data evidenced specific changes reflecting a difference in terms of metabolite concentration and taxa abundance in the pairwise statistical comparisons.

Further research is needed to better understand the mechanisms involved and explore the potential interventions targeting these pathways.

## Figures and Tables

**Figure 1 nutrients-17-00072-f001:**
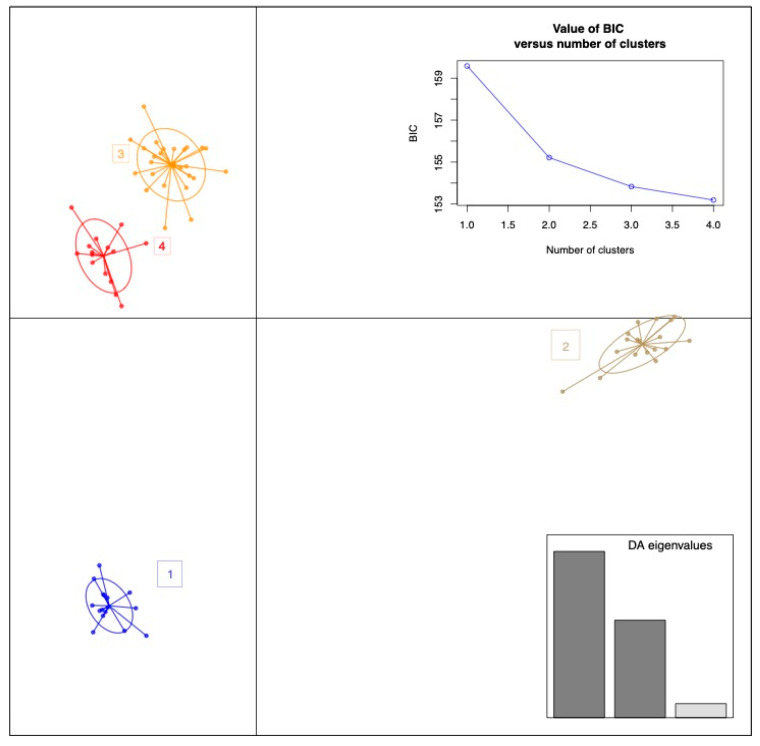
A priori group stratification resulting from the DAPC analysis run using the clinical/biochemical and anthropometric complete parameter matrix obtained from the 58-patient set. Used eigen values have been colored in dark grey.

**Figure 2 nutrients-17-00072-f002:**
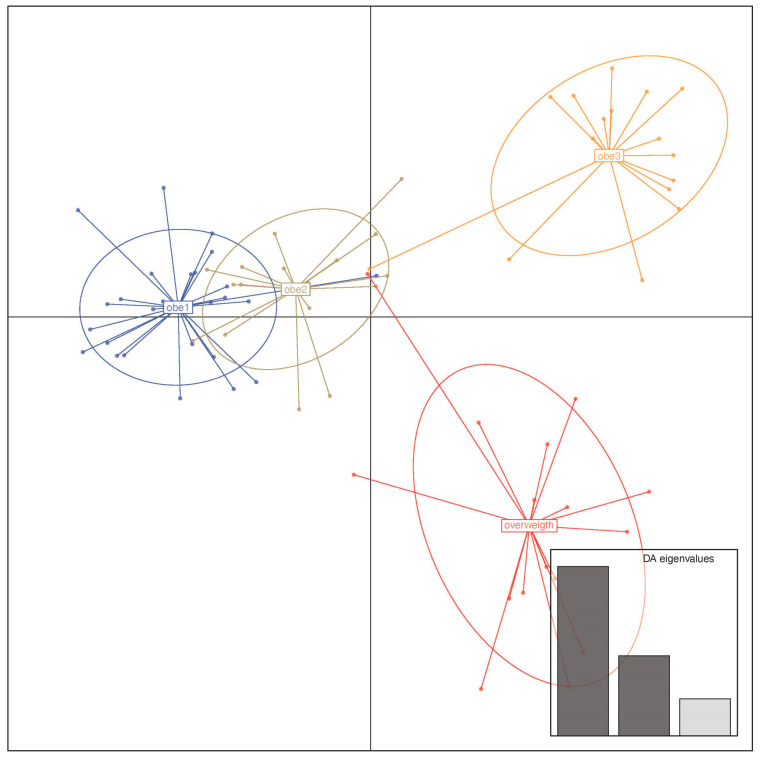
Clinical/anthropometric a posterior sample stratification in the DAPC analysis. The a posterior group assignment was based on BMI grouping, such as overweight, type 1, 2, and type 3 obesity.

**Figure 3 nutrients-17-00072-f003:**
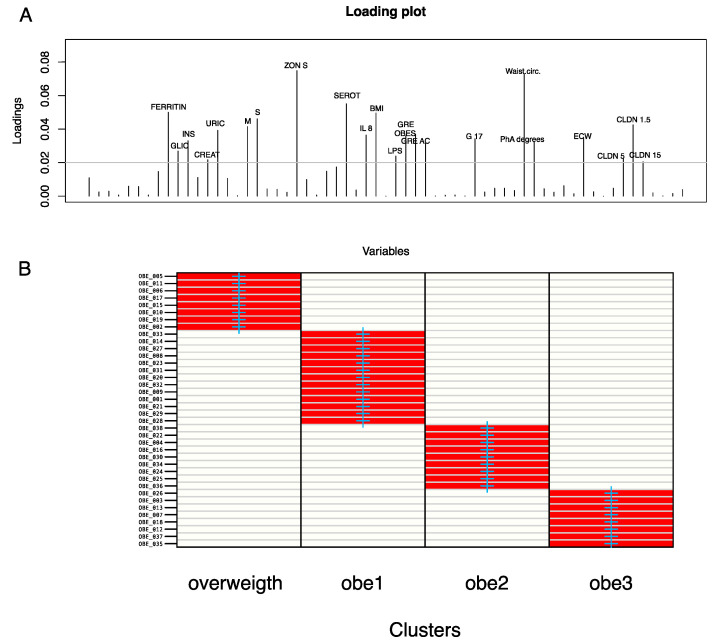
DAPC loading and assignment plot based on the 58-patient sample’s clinical/biochemical and anthropometric parameters. (**A**) DAPC loading plot reporting the clinical/anthropometric variables that most impacted cluster separation. An arbitrary 0.02 threshold is used to show the above threshold variables. (**B**) The cell matrix reports the fitting between the “a priori” and the “a posterior” assignments.

**Figure 4 nutrients-17-00072-f004:**
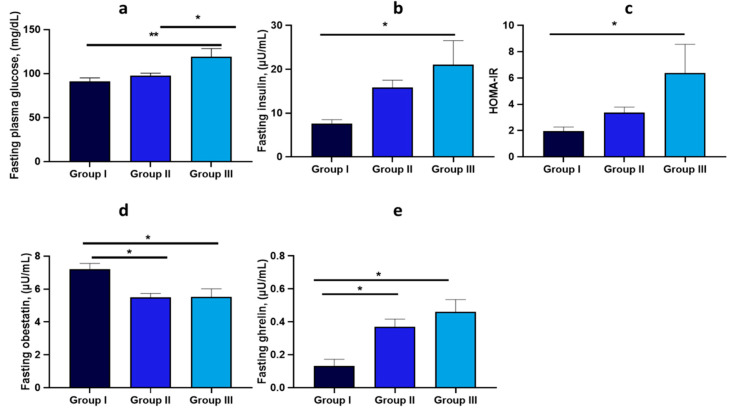
Comparison of glucose metabolism and endocrine indicators based on DAPC BMI stratification in overweight and obese subjects. Data are expressed as the mean ± standard deviation, and statistical analyses were performed using an ANOVA followed by a Tukey’s post hoc test. Statistically significant comparisons (*p* < 0.05) are highlighted by a bold line and marked with an asterisk. Path coefficients and significance: * *p* < 0.05, ** *p* < 0.01. Measured parameters include (**a**) fasting plasma glucose, (**b**) fasting insulin, (**c**) HOMA-IR, (**d**) fasting obestatin, (**e**) fasting ghrelin.

**Figure 5 nutrients-17-00072-f005:**
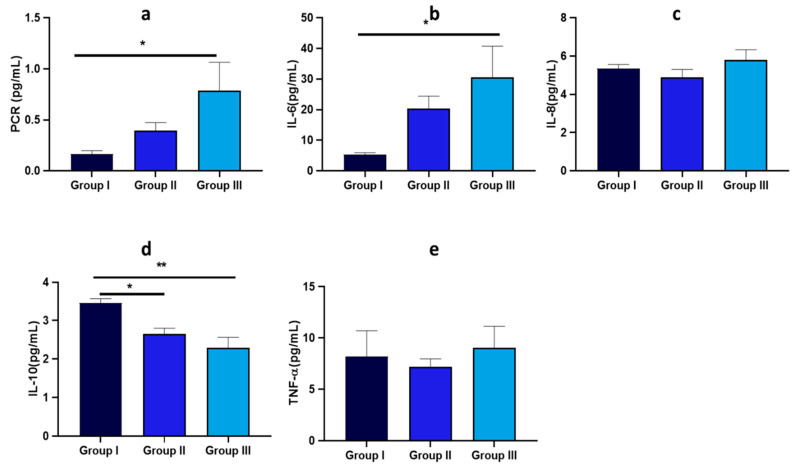
Levels of inflammatory markers in the 58 overweight and obese individuals grouped according to BMI categories. Data are expressed as the mean ± standard deviation, and statistical analyses were performed using an ANOVA followed by a Tukey’s post hoc test. Statistically significant comparisons (*p* < 0.05) are highlighted by a bold line and marked with an asterisk. Path coefficients and significance: * *p* < 0.05, ** *p* < 0.01. Inflammatory marker sub-panels include (**a**) PCR, (**b**) IL-6, (**c**) IL-8, (**d**) IL-10, (**e**) TNF-alpha.

**Figure 6 nutrients-17-00072-f006:**
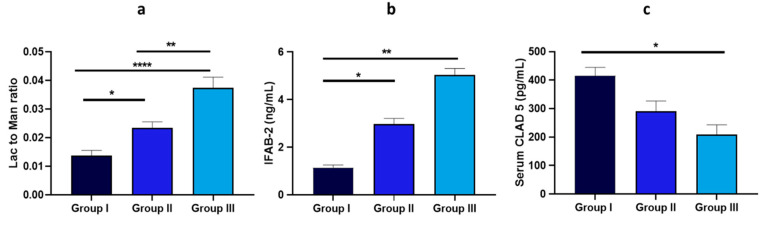
Levels of biomarkers related to intestinal barrier function and integrity measured in the set composed of 58 patients. Data are expressed as the mean ± standard deviation, and statistical analyses were performed using an ANOVA followed by a Tukey’s post hoc test. Statistically significant comparisons (*p* < 0.05) are highlighted by a bold line and marked with an asterisk. Path coefficients and significance: * *p* < 0.05, ** *p* < 0.01, **** *p* < 0.0001. Sub-panels include (**a**) lac/man ration, IFAB-2 (**b**), (**c**) serum claudin 5.

**Figure 7 nutrients-17-00072-f007:**
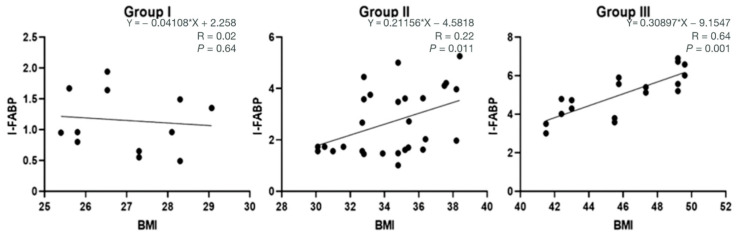
Linear regression analysis assessing the relationship between BMI and the intestinal permeability marker I-FABP.

**Figure 8 nutrients-17-00072-f008:**
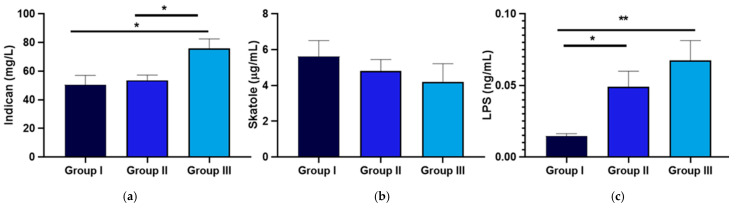
Levels of urinary indole, urinary skatole, and serum lipopolysaccharide (LPS) in the study cohort where the 58 patients have been grouped based on the DAPC BMI clusters. Data are expressed as the mean ± standard deviation, and statistical analyses were performed using an ANOVA followed by a Tukey’s post hoc test. *p*-values indicating significant differences (*p* < 0.05) are highlighted by a bold line and marked with an asterisk. Path coefficients and significance: * *p* < 0.05, ** *p* < 0.01. The cut-off levels indicating dysbiosis were set at 20 mg/L for indican and 20 μg/L for skatole. Sub-panels of urinary markers include (**a**) indican, (**b**) skatole and, (**c**) LPS.

**Figure 9 nutrients-17-00072-f009:**
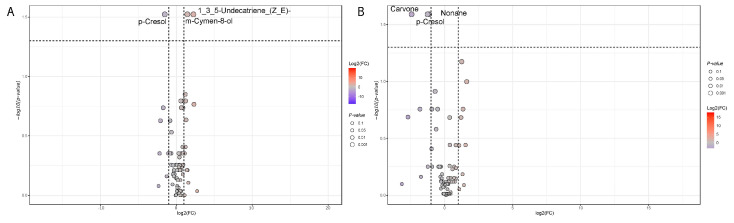
Statistically significant urinary VOCs detected by metabolomic (GC/MS) analyses on 37 patient samples. Fold change analysis was joined with a Welch’s corrected test (BH multiple correction) based on taxa at the genus level. A dot size increase is representative of lower *p*-values. Log2(FC) values range from gray (lower) to red (higher). Increased and decreased VOC concentrations are relative to the first comparison member, i.e., Group II versus Group I (**A**). (**B**) Pairwise comparison between Group III and Group I samples.

**Figure 10 nutrients-17-00072-f010:**
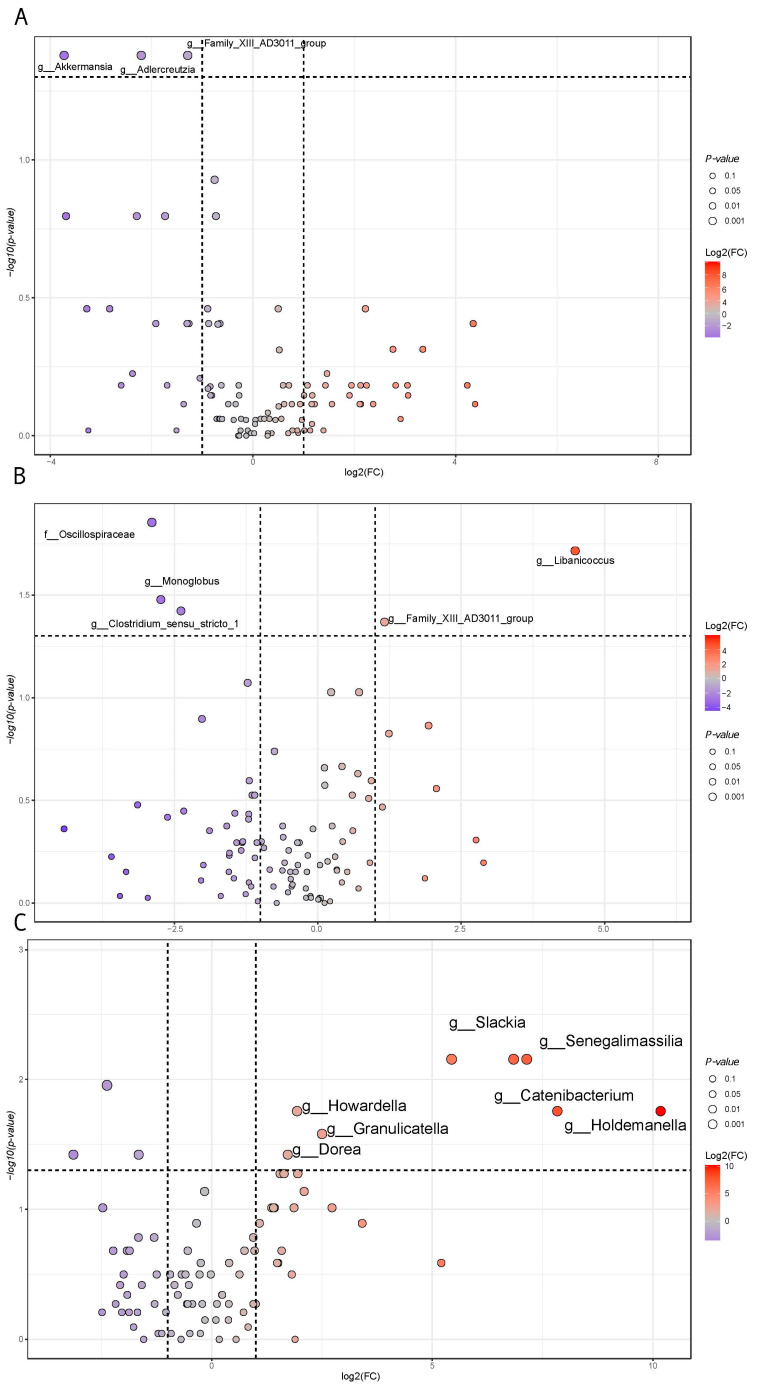
Statistically significant taxa volcano plot. Fold change analysis was joined with a Welch’s corrected test (BH multiple correction) based on taxa at the genus level. A dot size increase is representative of lower *p*-values. Log2(FC) values range from gray (lower) to red (higher). Increased and decreased VOC concentrations are relative to the first comparison member, i.e., Group II (**A**) versus Group I (**B**) pairwise comparison between Group III and Group II samples. (**C**) Comparison between Group II and Group I samples.

**Figure 11 nutrients-17-00072-f011:**
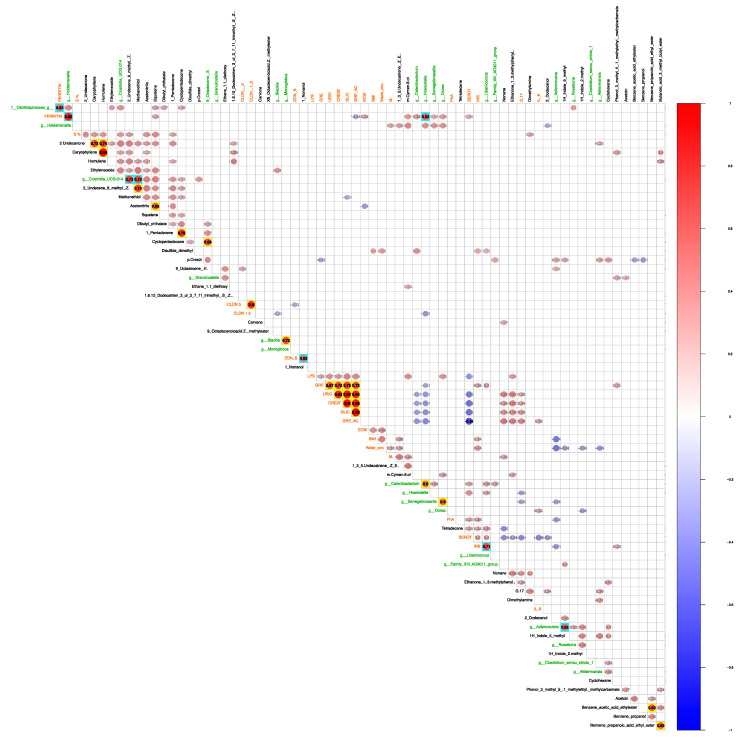
Pearson’s correlations among the VOC, taxa, and clinical variables. Statistically significant VOC (black), clinical/anthropometrical (dark orange), and taxa (dark green) variable sets have been correlated via a Pearson’s test. Only inter-group variable correlations with a *p*-value equal/lower than 0.05 have been shown, and only correlations greater than 0.6 were flagged in bold black font. Positive and negative correlations were reported as red and blue bubbles, respectively. Based on inter- and intra-group variable comparison (taxa, VOC, and clinical variables), bubbles were placed on a light aqua or yellow background.

**Table 1 nutrients-17-00072-t001:** Demographic, anthropometric, and bioimpedance features of the study groups (58 subjects) based on body mass index (BMI).

	Group I	Group II	Group III
	Mean ± SD	Mean ± SD	Mean ± SD
Age	45.75 ± 13.10	43.43 ± 15.28	37.75 ± 11.09
Weight (kg)	78.44 ± 7.835	94.84 ± 12.59 *	118.6 ± 24.60 ****^, ##^
Height (cm)	170.3 ± 8.102	168.9 ± 8.175	165.4 ± 14.85
BMI (kg/m^2^)	27.01 ± 1.386	34.15 ± 2.616 ****	45.54 ± 3.065 ****^, ####^
Waist Circumference (cm)	95.63 ± 6.865	109.5 ± 13.74 *	128.3 ± 10.05 ****^, ##^
Fat Mass (kg)	26.53 ± 3.728	40.29 ± 7.363 ***	53.54 ± 5.737 ****^, ###^
Fat-Free Mass (kg)	51.9 ± 8.293	59.98 ± 9.793	70.88 ± 20.92 *
Phase Angle (°)	6.238 ± 0.8501	12.59 ± 18.45	13.4 ± 19.25
Total Body Water (kg)	38.43 ± 5.988	44.53 ± 6.927	55.81 ± 15.36 *^, ##^
Extracellular Body Water (kg)	16.91 ± 1.936	23.03 ± 8.934	28.63 ± 4.124 **

Data are presented as the mean ± standard deviation, with statistical analysis conducted by means of an ANOVA followed by a Tukey’s post hoc test. Statistically significant differences (*p* < 0.05) are highlighted by an asterisk if emerging from the comparison of both Groups II and III versus Group I and by a pound symbol in the case that significance is derived from comparing Group II and Group III. Path coefficients and significance: * *p* < 0.05, **^, ##^
*p* < 0.01, ***^, ###^
*p* < 0.001, ****^, ####^
*p* < 0.0001.

**Table 2 nutrients-17-00072-t002:** Blood clinical parameters of the 58 patients based on the DAPC BMI grouping.

	Group I	Group II	Group III
	Mean ± SD	Mean ± SD	Mean ± SD
Total Cholesterol (mg/dL)	211.4 ± 39.71	198.1 ± 26.68	225.7 ± 47.58
HDL Cholesterol (mg/dL)	53 ± 15.91	58.79 ± 14.06	44.97 ± 13.94
LDL Cholesterol (mg/dL)	141.1 ± 39.68	132.1 ± 22.70	159 ± 41.74
Triglycerides (mg/dL)	69.29 ± 20.44	89 ± 28.97	205.6 ± 79.62 ****^, ####^
25-OH-Vitamin D (ng/mL)	24.18 ± 11.58	22.86 ± 5.389	13.9 ± 4.463 *^, #^
Iron (µg/dL)	100.6 ± 47.27	78.54 ± 27.37	96.9 ± 28.94
Ferritin (ng/mL)	82.3 ± 90.13	136.1 ± 182.6	139.7 ±110.4
Uric Acid (mg/dL)	5.113 ± 1.815	5.578 ± 1.436	5.717 ± 0.2927
Creatinine (mg/dL)	0.7213 ± 0.1009	0.7892 ± 0.1211	0.775 ± 0.1865

Data are presented as the mean ± standard deviation, with statistical analysis conducted using an ANOVA followed by a Tukey’s post hoc test. Statistically significant differences (*p* < 0.05) are highlighted by an asterisk if emerging from the comparison of both Groups II and III versus Group I and by a pound symbol in the case that significance is derived from comparing Group II and Group III. Path coefficients and significance: *^, #^
*p* < 0.05, ****^, ####^
*p* < 0.0001.

**Table 3 nutrients-17-00072-t003:** Fecal metabolites from 37 samples inspected by means of a fold change analysis joined with a Welch’s test: the list of statistically significant volatile organic compounds (VOCs) in pairwise comparisons.

Group III vs. Group II	FC	log2(FC)	raw.pval	−LOG10(p)
Caryophyllene	0.28543	−1.8088	0.0035686	2.4475
Dibutylphthalate	5.21 × 10^−5^	−14.229	0.030786	1.5116
2 Undecene 6 methyl(Z)	7.11 × 10^−5^	−13.779	0.034319	1.4645
Cyclohexane	0.17531	−2.512	0.036752	1.4347
Squalene	5.90 × 10^−5^	−14.048	0.038522	1.4143
Benzene propanoic acid ethylester	0.16315	−2.6157	0.046753	1.3302
Acetoin	0.022938	−5.4461	0.049827	1.3025
**Group II vs. Group I**	**FC**	**log2(FC)**	**raw.pval**	**−LOG10(p)**
Tetradecane	0.088078	−3.5051	0.0012218	2.913
Ethane 1,1 diethoxy	7.6345	2.9325	0.0029338	2.5326
1H Indole 2 methyl	0.15457	−2.6937	0.0058608	2.232
Dimethylamine	0.068047	−3.8773	0.010333	1.9858
2 Dodecanol	0.30123	−1.7311	0.019857	1.7021
1H Indole 5,methyl	0.1325	−2.9159	0.031315	1.5042
Dimethylsulfide	0.099079	−3.3353	0.040285	1.3949
9 Octadecene (E)	0.35901	−1.4779	0.042605	1.3705
Humulene	14151	13.789	0.042715	1.3694
**Group III vs. Group I**	**FC**	**log2(FC)**	**raw.pval**	**−LOG10(p)**
Tetradecane	1.37 × 10^−5^	−16.16	0.0010575	2.9757
1H Indole 2 methyl	1.14 × 10^−8^	−26.386	0.0022662	2.6447
Squalene	2.14 × 10^−7^	−22.158	0.0074536	2.1276
Ethanone 1 (3 aminophenyl)	0.029422	−5.087	0.0091548	2.0384
Phenol 3 methyl 5(1 methylethyl) methylcarbamate	0.12261	−3.0279	0.013715	1.8628
Cyclohexane	0.14551	−2.7808	0.015555	1.8081
Benzenepropanol	6.3568	2.6683	0.016863	1.7731
Methanethiol	0.00025876	−11.916	0.020583	1.6865
2 Undecanone	0.041823	−4.5796	0.024052	1.6188
1H Indole 5 methyl	0.32174	−1.6361	0.025754	1.5892
1 Pentadecene	0.075389	−3.7295	0.026016	1.5848
9 Octadecenoicacid(Z) methylester	0.090349	−3.4683	0.026327	1.5796
1 Nonanol	0.062512	−3.9997	0.027971	1.5533
Butanoicacid 3 methylbutyl ester	53850	15.717	0.030967	1.5091
Benzeneaceticacid ethyl ester	3.1594	1.6596	0.032752	1.4848
Acetoin	0.026044	−5.2629	0.037233	1.4291
Cyclopentadecane	0.05617	−4.1541	0.038908	1.41
Disulfide dimethyl	0.22422	−2.157	0.040839	1.3889
Dimethylsulfide	2.34 × 10^−5^	−15.382	0.041756	1.3793
1 6 10 Dodecatrien 3 ol 3 7 11 trimethyl [S (Z)]	0.00014158	−12.786	0.041756	1.3793
2 Undecene 6 methyl (Z)	8.18 × 10^−6^	−16.899	0.043923	1.3573
Ethyleneoxide	2.62 × 10^−5^	−15.218	0.045674	1.3403
2 Dodecanol	0.14026	−2.8339	0.046023	1.337
Acetonitrile	3.21 × 10^−6^	−18.247	0.047046	1.3275

All reported hits have a *p*-value lower than 0.05.

## Data Availability

The datasets used and/or analyzed during the current study are available from the corresponding author upon reasonable request.

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
