# Peer review of "Role of Increasing Body Mass Index in Gut Barrier Dysfunction, Systemic Inflammation, and Metabolic Dysregulation in Obesity"

_nutrients, 2024, doi:10.3390/nu17010072_

Round 1

Reviewer 1 Report

Comments and Suggestions for Authors

Abstract: please provide in the results, for example, how many people completed the study, i.e. 37 people, the most important results with the p value. The results in the abstract are written too generally.

Methods:

- The sentence "The research, a longitudinal study in the already explored cohort, focused on analyzing data from obese participants aged from 18 to 65 years, all with a BMI above 25 kg/m²." Maybe it would be worth changing "obese participants" to "participants with excessive body mass"? because you included people with obesity, but also overweight.

- "2.2. Anthropometric Assessment" It is worth adding in the content of this point the literature source on the basis of which you calculated BMI and WC. As well as providing categories/interpretations of these 2 parameters. In the case of bMI, you provided categories in point 3.1 (in my opinion it should be in methods). In the case of WC, maybe some guidelines giving instructions for measuring "For waist circumference (WC), patients were asked to stand upright with their abdomen exposed and weight evenly distributed on both legs. The tape measure was positioned between the lower edge of the ribs and the iliac crest, approximately at the navel level."

- point 2.3: Fragment "All patients were examined according to the European Society of Parenteral and Enteral Nutrition (ESPEN) recommendations. " please cite the literature source.

Results

- fragment "Next, we categorized the sample into four groups based on BMI (overweight and three obesity classes) and we used this grouping in a subsequent DAPC analysis (Figure 2)." I disagree with "Based on BMI, the cohort was further divided into three sub-groups: Group I (BMI 25-29.9), Group II (BMI 30-39.9), and Group III (BMI > 40)." do we have 3 or 4 groups in this study?

- it is worth writing how many people were in which group in each table.

Discussion

The discussion consists of comparing the results of our study with other studies. In connection with this, I have suggestions for some paragraphs:

- unnecessary paragraph "Our longitudinal cohort study aimed to explore the link between increasing BMI and gut integrity, inflammatory markers, lipid profiles, gut microbiota and volatile metabolites, providing insights into how obesity may compromise the gut barrier and contribute to metabolic disorders as BMI values ​​grow. Based on DAPC clustering stratification, we divided our cohort into three BMI groups (25-29.9, 30-39.9, >40). Gut microbiota and metabolite profiles were then analyzed using metataxonomics and GC/MS untargeted metabolomics."

- no literature reference "The intestinal dysbiosis observed in obese subjects, quantified by increased pro-inflammatory bacteria and reduced beneficial bacteria (such as Roseburia and Adlercreutzia), results in decreased production of protective metabolites like butyrate and indole compounds."

- no reference to literature "Inflammatory markers showed a marked difference in Group III, with significantly higher levels of pro-inflammatory markers like PCR and IL-6 and lower levels of the anti-inflammatory marker IL-10. TNF and IL-8 levels, however, remained similar across the groups."

- no reference to literature "Gut barrier function and integrity markers, such as the Lactulose/Mannitol absorption ratio, increased significantly in Groups II and III compared to Group I and were also higher in Group III compared to Group II. Serum I-FABP levels followed a similar trend, while Claudin 5 levels progressively decreased as BMI increased. No significant differences were observed in serum and fecal zonulin or Claudin 15 levels."

Conclusions should be written in the conclusions, not in the discussion. At the end of the discussion, please write the strengths and limitations of your study.

Author Response

Reviewer 1

Abstract: please provide in the results, for example, how many people completed the study, i.e. 37 people, the most important results with the p value. The results in the abstract are written too generally.

We thank the reviewer for her/his useful and precious comments. We extensively modified the abstract body, and we better specified the obtained results.

Methods:

- The sentence "The research, a longitudinal study in the already explored cohort, focused on analyzing data from obese participants aged from 18 to 65 years, all with a BMI above 25 kg/m²." Maybe it would be worth changing "obese participants" to "participants with excessive body mass"? because you included people with obesity, but also overweight.

Thank you. We perfectly agree with the reviewer comment and contextually modified the text.

- "2.2. Anthropometric Assessment" It is worth adding in the content of this point the literature source on the basis of which you calculated BMI and WC. As well as providing categories/interpretations of these 2 parameters. In the case of bMI, you provided categories in point 3.1 (in my opinion it should be in methods). In the case of WC, maybe some guidelines giving instructions for measuring "For waist circumference (WC), patients were asked to stand upright with their abdomen exposed and weight evenly distributed on both legs. The tape measure was positioned between the lower edge of the ribs and the iliac crest, approximately at the navel level."

Many thanks. We moved the BMI description in the methods as suggested. We added the source from where we take the useful guidelines for WC and BMI as reported by World Health Organization (https://iris.who.int/bitstream/handle/10665/44583/9789241501491_eng.pdf)

- point 2.3: Fragment "All patients were examined according to the European Society of Parenteral and Enteral Nutrition (ESPEN) recommendations. " please cite the literature source.

Many thanks. We added the http link in the text (https://www.gavecelt.it/nuovo/links-utili/espen-european-society-parenteral-and-enteral-nutrition)

Results

- fragment "Next, we categorized the sample into four groups based on BMI (overweight and three obesity classes) and we used this grouping in a subsequent DAPC analysis (Figure 2)." I disagree with "Based on BMI, the cohort was further divided into three sub-groups: Group I (BMI 25-29.9), Group II (BMI 30-39.9), and Group III (BMI > 40)." do we have 3 or 4 groups in this study?

We thank the reviewer for this precious comment that offers us the possibility to better clarify that the a priori clustering analysis identified 4 different groups but considering that patients with Obesity I and Obesity II almost overlapped in the DAPC plot we considered them as a unique group.

- it is worth writing how many people were in which group in each table.

Many thanks for this required modification. We added the requested information in each table.

Discussion

The discussion consists of comparing the results of our study with other studies. In connection with this, I have suggestions for some paragraphs:

- unnecessary paragraph "Our longitudinal cohort study aimed to explore the link between increasing BMI and gut integrity, inflammatory markers, lipid profiles, gut microbiota and volatile metabolites, providing insights into how obesity may compromise the gut barrier and contribute to metabolic disorders as BMI values grow. Based on DAPC clustering stratification, we divided our cohort into three BMI groups (25-29.9, 30-39.9, >40). Gut microbiota and metabolite profiles were then analyzed using metataxonomics and GC/MS untargeted metabolomics."

Many thanks. As suggested, we removed the paragraph.

- no literature reference "The intestinal dysbiosis observed in obese subjects, quantified by increased pro-inflammatory bacteria and reduced beneficial bacteria (such as Roseburia and Adlercreutzia), results in decreased production of protective metabolites like butyrate and indole compounds."

Many thanks. The result presented in this sentence has been corroborated by the reference reported also in the subsequent paragraph.

- no reference to literature "Inflammatory markers showed a marked difference in Group III, with significantly higher levels of pro-inflammatory markers like PCR and IL-6 and lower levels of the anti-inflammatory marker IL-10. TNF and IL-8 levels, however, remained similar across the groups."

Many thanks. We better explained that this sentence refers to our results. We anyway cited a reference in line with them, dealing with inflammation (see ref 4, 13 and 40)

- no reference to literature "Gut barrier function and integrity markers, such as the Lactulose/Mannitol absorption ratio, increased significantly in Groups II and III compared to Group I and were also higher in Group III compared to Group II. Serum I-FABP levels followed a similar trend, while Claudin 5 levels progressively decreased as BMI increased. No significant differences were observed in serum and fecal zonulin or Claudin 15 levels."

Many thanks for this comment. We added the reference as requested (See page 21, from line 634 to line 641, see refs 40,41 and 42).

Conclusions should be written in the conclusions, not in the discussion. At the end of the discussion, please write the strengths and limitations of your study.

Many thanks. We moved the conclusions in the proper sentence. We added a paragraph with strengths and limitations.

Reviewer 2 Report

Comments and Suggestions for Authors

The manuscript written by Maqoud et al is a comprehensive study aimed at understanding the relationship between body weight and the intestinal barrier. The text is clearly written and the topic is relevant. However, the quality could be improved. In addition, it is very difficult to read the take-home messages of the study.  It is also not clear from the text why VOCs were measured.

Minor comments:

The quality of figure 1 needs to be improved.

Figure 2 is strange. Please revise it.

Bottom of page 8: some words are larger than others.

Figure 3: I cannot see the orange panel. Why have you defined this colour code in the legend?

Legend to Figure 3: The size of the letters is different.

Table 2: In which matrix were the characteristics measured? The parameters in the first rows are typical for plasma, but at the end are typical for urine.

Section 3.7: Can you at least show the data in the appendix?

Correct letter type at the end of section 3.10.

Comments on the Quality of English Language

Whether the text is grammatically correct or not, I do not feel competent to judge.

Author Response

Reviewer 2

The manuscript written by Maqoud et al is a comprehensive study aimed at understanding the relationship between body weight and the intestinal barrier. The text is clearly written and the topic is relevant. However, the quality could be improved. In addition, it is very difficult to read the take-home messages of the study. It is also not clear from the text why VOCs were measured.

We thank the reviewer for her/his precious comments. We extensively modify the body text and improved the workflow. We better centred the topic on VOCs and have extended our discussion and added citations in line with this aspect (See page 21, from line 650 to line 662, and from line 671 to line 675, see refs 43, 44, 45,46, 47 and 48).

Minor comments:

The quality of figure 1 needs to be improved.

Figure 1 has been revised and a new pdf version with higher pixel was prepared.

Figure 2 is strange. Please revise it.

Figure 2 reports DAPC plot in which only Eigen values and cluster are usually reported. The axes do not have any graduated scale as the output as been provided in this way by the R package.

Bottom of page 8: some words are larger than others.

Many thanks. We made the required modifications.

Figure 3: I cannot see the orange panel. Why have you defined this colour code in the legend?

Many thanks. That was a typo in the legened. The orange and yellows refer to a graduated scale that in our case was not present.

Legend to Figure 3: The size of the letters is different.

Many thanks, we corrected the typo.

Table 2: In which matrix were the characteristics measured? The parameters in the first rows are typical for plasma, but at the end are typical for urine.

All the parameters reported are from blood. We specified this information in the legend.

Section 3.7: Can you at least show the data in the appendix?

Many thanks. These data are relative to intestinal barrier integrity and function and both the figures are important to show the statistically significance of measured variables relative to intestinal permeability (IFABP) and the correlation with BMI (regression analysis). If possible, we would like to keep these part in the main text.

Correct letter type at the end of section 3.10.

Many thanks. Corrected.
